



# Aerosol pH and liquid water content determine when particulate matter is sensitive to ammonia and nitrate availability

Athanasios Nenes[1,2*], Spyros N. Pandis[1,3], Rodney J. Weber[4], Armistead Russell[5]

[1]Institute for Chemical Engineering Sciences, Foundation for Research and Technology Hellas,
Patras, GR-26504, Greece
[2]School of Architecture, Civil & Environmental Engineering, Ecole polytechnique fédérale de
    Lausanne, CH-1015, Lausanne, Switzerland
[3]Department of Chemical Engineering, University of Patras, GR-26504, Greece
[4]School of Earth and Atmospheric Sciences, Georgia Institute of Technology, Atlanta, GA 30332,
USA
[5]School of Civil & Environmental Engineering, Georgia Institute of Technology, Atlanta, GA
    30332, USA

*   correspondence to athanasios.nenes@epfl.ch

**Abstract.** Nitrogen oxides ($NO_x$) and ammonia ($NH_3$) from anthropogenic and biogenic emissions are central contributors to particulate matter (PM) concentrations worldwide. The response of PM to changes in the emissions of both compounds is typically studied on a case-by-case basis, owing in part to the complex thermodynamic interactions of these aerosol precursors with other PM

constituents. Here we present a simple but thermodynamically consistent approach that expresses the chemical domains of sensitivity of aerosol particulate matter to $NH_3$ and $HNO_3$ availability in terms of aerosol pH and liquid water content. From our analysis, four policy-relevant regimes emerge in terms of sensitivity: *i*) $NH_3$-dominated, *ii*) $HNO_3$-dominated, *iii*) combined $NH_3$ and $HNO_3$ sensitive, and, *iv*) a domain where neither $NH_3$ and $HNO_3$ are important for PM levels (but

only nonvolatile precursors such as NVCs and sulfate). When this framework is applied to ambient measurements or predictions of PM and gaseous precursors, the "chemical regime" of PM sensitivity to $NH_3$ and $HNO_3$ availability is directly determined. The use of these regimes allows novel insights and is an important tool to evaluate chemical transport models. With this extended understanding, aerosol pH and associated liquid water content naturally emerge as previously

ignored state parameters that drive PM formation.



## 1. Introduction

Gas-phase ammonia ($NH_{3(g)}$, hereon "$NH_3$") is one of the most important atmospheric alkaline species and contributor to atmospheric fine particle mass (Seinfeld and Pandis, 2016). $NH_3$ originates from nitrogen-based fertilizer, animal waste (e.g., Aneja et al., 2009), biomass burning

(e.g., Behera et al., 2013) and the natural biosphere (NRC, 2016). $NH_3$ emissions are also linked to world food production, so these are expected to increase with world population (NRC, 2016). Ammonia reacts with sulfuric and nitric acids (from $SO_2$ and $NO_x$ oxidation) to form ammonium sulfate/bisulfate and nitrate aerosol that globally constitute an important fraction of ambient $PM_{2.5}$ mass (Kanakidou et al., 2005; Sardar et al., 2005; Zhang et al., 2007). $SO_2$ and $NO_x$ emissions are

expected to decrease over time due to air quality regulations (IPCC, 2013). Combined with increasing $NH_3$ levels (e.g., Skjøth and Geels, 2013), this may lead to changes in aerosol composition and mass concentration, with important impacts on human health (Pope et al., 2004; Lim et al., 2012; Lelieveld et al., 2015; Cohen et al., 2017), ecosystem productivity (Fowler et al., 2013) and the climate system (Haywood and Boucher, 2000; Bellouin et al., 2011; IPCC, 2013).

The above emissions trends have created the expectation that atmospheric aerosol will become significantly less acidic over time (West et al., 1999; Pinder et al., 2007, 2008; Heald et al., 2012; Tsimpidi et al., 2007; Saylor et al., 2015). Reductions in ammonium sulfate due to $SO_2$ reductions can be balanced, at least in part, by ammonium nitrate formation (e.g., West et al., 1999; Heald et al., 2014; Karydis et al., 2016; Vasilakos et al., 2018). This behavior arises because nitrate may

remain in the gas phase as $HNO_3$ when insufficient amounts of total ammonia (i.e., gas+aerosol) or non-volatile cations (NVCs) from dust and seasalt exist to "neutralize" aerosol sulfate (i.e., completely consume any free sulfuric acid or bisulfate salts). This conceptual model can fail, because it does not sufficiently consider the large volatility difference between deliquesced aerosol containing sulfate/NVCs and ammonium/nitrate the latter two of which is strongly modulated by

aerosol acidity (pH) (Guo et al., 2015; Weber et al., 2016; Guo et al., 2017) and changes in the uptake of water due to compositional change. Modeling studies explicitly considering acidity effects may still incorrectly estimate nitrate substitution, owing to errors in emissions of non-volatile cations (such as Na, Ca, K and Mg) that can bias aerosol acidity and ammonium or nitrate partitioning (Vasilakos et al., 2018). A bias in our understanding of aerosol pH can reaffirm a

sometimes incorrect conceptual model of aerosol nitrate formation, and fundamental reasons for



prediction biases in nitrate and ammonium (i.e., errors in pH and liquid water content) are not identified – therefore inhibiting further model improvement.

Developing an understanding of when aerosol levels are sensitive to $NH_3$ and $HNO_3$ concentrations requires a new approach that explicitly considers aerosol pH and its effects on aerosol precursor

volatility in a thermodynamically consistent way. Here we present such a framework, and demonstrate it with observational data to understand the "chemical regimes" associated with aerosol sensitivity to changes in ammonia and nitrate availability.

## 2. The new conceptual framework

Aerosol pH needs to be sufficiently high for aerosol nitrate formation to readily occur. Depending

on the temperature and the amount of liquid water this threshold ranges between a pH of 1.5 and 3.5 (Meskhidze et al., 2003; Guo et al., 2016, 2017; Fig. 1). If pH is high enough, almost all inorganic nitrate forming from $NO_x$ oxidation mostly resides in the aerosol phase; when pH however is low (typically below 1.5 to 2), nitrate remains almost exclusively in the gas phase as $HNO_3$, regardless of the amount present. Between these "high" and "low" pH values, a "sensitivity

window" emerges, where partitioning shifts from nitrate being predominantly gaseous to mostly aerosol-bound. When acidity is below this "pH window", aerosol nitrate is almost nonexistent and therefore aerosol levels are insensitive to $HNO_3$ availability and controls aimed solely on aerosol nitrate reduction are unimportant since none is in the aerosol phase. When the pH is above the window, most nitrate resides in the aerosol phase, and aerosol levels directly respond to $HNO_3$

availability. A similar situation exists for aerosol ammonium – although with an inverse dependence on pH, compared to $HNO_3$. When aerosol pH is low enough any inorganic ammonia emitted mostly resides in the aerosol phase – and when pH is high enough, most of it resides in the gas phase (Fig. 1). Based on the above, one can then define characteristic levels of aerosol acidity, where aerosol becomes insensitive to $NH_3$ (or $HNO_3$) concentrations, and vice versa. In the

following sections, we quantitatively develop these concepts and formulate a new thermodynamically consistent conceptual framework of aerosol formation.





## 2.1 Determining when aerosol mass is sensitive to nitric acid and ammonia availability

For a given airmass with total nitrate $NO_3^T$ (i.e., the amount of aerosol and gas-phase nitrate), the equilibrium aerosol nitrate concentration, $NO_3^-$, is given by $NO_3^- = \varepsilon(NO_3^-) \, NO_3^T$, where $\varepsilon(NO_3^-)$ is the fraction of $NO_3^T$ that partitions to the particle phase. Given that when nitrate ion partitions

to the aerosol, it is associated with semi-volatile $NH_4^+$ and nonvolatile cations (NVC) such as $Na^+$, $Ca^{+2}$, $K^+$ and $Mg^{+2}$, the sensitivity of aerosol mass to changes in $NO_3^T$ is proportional to the changes occurring in $NO_3^-$. Therefore:

$$\frac{\partial PM}{\partial NO_3^T} = \kappa \frac{\partial NO_3^-}{\partial NO_3^T} = \kappa\varepsilon(NO_3^-) \tag{1}$$

where $\kappa$ is the ratio of PM mass formed (or lost) per mol of $NO_3^-$ that condenses (or evaporates) from the particles. Therefore, if $NO_3^-$ is associated with aerosol $NH_4NO_3$, then $\kappa = 80/62=1.29$.

Lower values are found for particles rich in NVC that are associated with carbonates and chlorides; for example, if nitric acid were replacing chloride in seasalt (e.g., conversion of NaCl to NaNO3), the ratio would be $\kappa = (85-58.4)/62=0.43$. A similar $\kappa$ is seen when alkaline dust particles rich in CaCO3 react with HNO3 to form Ca(NO3)2, $\kappa=(164-100)/(2\times62)=0.51$. The sensitivity of PM to changes in $NO_3^T$ can therefore be expressed in terms of nitrate partitioning, so the parameters that

affect $\varepsilon(NO_3^-)$ also directly impacts $\frac{\partial PM}{\partial NO_3^T}$. We now proceed with explicitly quantifying how aerosol liquid water and pH affect nitrate partitioning, hence PM sensitivity to nitrate availability.

Meskhidze et al. (2003) and later on Guo et al. (2017) showed that for a deliquesced aerosol $\varepsilon(NO_3^-)$ explicitly depends on the concentration of $H^+$ in the aerosol phase, $[H^+]$, and the aerosol liquid water content, $W_i$, as:

$$\varepsilon(NO_3^-) = \frac{K_{n1}H_{HNO_3}W_iRT}{\gamma_{H^+}\gamma_{NO_3^-}[H^+] + K_{n1}H_{HNO_3}W_iRT} \tag{2}$$

where $H_{HNO_3}, K_{n1}$ is the Henry's law and acid dissociation constant for HNO3, respectively, $R$ is the universal gas constant, $T$ is the temperature and $\gamma_{H^+}, \gamma_{NO_3^-}$ are the single-ion activity coefficients for $H^+$, $NO_3^-$, respectively. Temperature dependence for $H_{HNO_3}$ is provided by Sander (2015), while activity coefficients can be calculated using an aerosol thermodynamic model (e.g., ISORROPIA-II; Fountoukis et al., 2007).





Similarly, equilibrium partitioning of $NH_3^T$ to the aerosol is given by $NH_4^+ = \varepsilon(NH_4^+)\, NH_3^T$, where $\varepsilon(NH_4^+)$ is the fraction of $NH_3^T$ (i.e., the amount of aerosol ammonium and gas-phase ammonia) that partitions to the particle phase. The sensitivity of aerosol mass to perturbations in total ammonia is $\frac{\partial PM}{\partial NH_3^T} = \lambda \frac{\partial NH_4^+}{\partial NH_3^T} = \lambda \varepsilon(NH_4^+)$, where $\lambda$ is the ratio of mass of PM that is lost/gained per

mol of evaporation/loss of $NH_4^+$. $\lambda$ is more variable than $\kappa$, because the anion associated with ammonium can be involatile or semi-volatile species with relatively large molar mass. For example, if $NH_4^+$ condenses/evaporates from sulfate salts ($NH_4HSO_4$, $(NH_4)_2SO_4$), then $\lambda = 18/17 = 1.06$, $\lambda = 4.4$ for $NH_4NO_3$ and $\lambda = 2.97$ for $NH_4Cl$.

From the above, the sensitivity of PM to changes in $NH_3^T$ can therefore be expressed in terms of

ammonium partitioning. $\varepsilon(NH_4^+)$, just like in Eq. 2, can be linked to aerosol liquid water and pH as (Guo et al., 2017):

$$\varepsilon(NH_4^+) = \frac{\frac{\gamma_{H^+}}{\gamma_{NH_4^+}}\frac{H_{NH_3}}{K_a}[H^+]W_i RT}{1 + \frac{\gamma_{H^+}}{\gamma_{NH_4^+}}\frac{H_{NH_3}}{K_a}[H^+]W_i RT} \tag{3}$$

where $H_{NH_3}, K_a$ is the Henry's law and dissociation constant for $NH_3$, respectively, and $\gamma_{NH_4^+}$ is the single-ion activity coefficient for $NH_4^+$, respectively. Temperature dependence for $H_{NH_3}$ is provided by Sander (2015).

Defining the parameters $\Psi = \frac{RTK_{n1}H_{HNO_3}}{\gamma_H + \gamma_{NO_3^-}}$ and $\Phi = \frac{\gamma_{H^+}}{\gamma_{NH_4^+}}\frac{H_{NH_3}}{K_a}RT$, equations (2) and (3) can be written as:

$$\varepsilon(NO_3^-) = \frac{\Psi W_i}{[H^+] + \Psi W_i}; \;\; \varepsilon(NH_4^+) = \frac{\Phi[H^+]W_i}{1 + \Phi[H^+]W_i} \tag{4}$$

For given levels of $W_i$, Equations (4) yield "sigmoidal" functions that display a characteristic "pH sensitivity window", where the partition fraction changes from zero to unity over a limited pH range. Equations (4) can then be used to determine a "characteristic pH" that defines when aerosol

is insensitive to total ammonia and nitrate availability (or emissions). For this purpose, we determine the pH for which $\varepsilon(NO_3^-)$ and $\varepsilon(NH_4^+)$ are equal to a characteristic (small) threshold value, being $\alpha$ for $\varepsilon(NO_3^-)$ and $\beta$ for $\varepsilon(NH_4^+)$ (Figure 1). When $\alpha$ (or $\beta$) are exceeded, the aerosol is said to be sensitive to $NH_3$ (or $NO_x$) emissions, because changes in $NH_3$, $NO_x$ levels can



appreciably affect aerosol concentrations. This sensitivity may be in one direction (e.g., increase of the emissions if the corresponding particulate levels are low and decrease if they are high) or in both. Guo et al. (2018) found a "critical" pH of approximately 3, above which the $\varepsilon(NO_3^-)$ is nearly 1 and almost all nitrate ($NO_3^T$) is in the gas phase ($HNO_3$). Here we generalize the approach

developing relationships between the terms that depend on aerosol composition, pH and particle water, with temperature still remaining as an independent variable.

Based on the above discussion, the characteristic acidity level for nitrate, $pH'$, is computed as

$$\alpha = \frac{\Psi W_i}{[H^+]' + \Psi W_i} \Longrightarrow [H^+]' = \frac{(1-\alpha)}{\alpha}\Psi W_i \Longrightarrow$$

$$pH' = -log\left[\left(\frac{1-\alpha}{\alpha}\right)\Psi W_i\right] \tag{5}$$

where $[H^+]'$ is the concentration where $\varepsilon(NO_3^-)$ equals the threshold value. The parameter $\left(\frac{1-\alpha}{\alpha}\right)$, which we call the "threshold factor", adjusts $pH'$ to account for the threshold above which the

aerosol is said to become sensitive to $NO_3^T$.

Similarly to nitrates, the characteristic acidity level for ammonium, $pH''$, is determined as

$$\beta = \frac{\Phi[H^+]W_i}{1 + \Phi[H^+]W_i} \Longrightarrow [H^+] = \frac{1}{\frac{(1-\beta)}{\beta}\Phi W_i} \Longrightarrow$$

$$pH'' = log\left[\left(\frac{1-\beta}{\beta}\right)\Phi W_i\right] \tag{6}$$

## 2.2 Chemical domains of aerosol mass sensitive to nitrate and ammonia perturbations

Hereon we consider $\alpha = \beta = 0.1$; in selecting these threshold values, we assume that aerosol responds in an important manner to $NH_3/HNO_3$ emissions when at least 10% of the total precursor

can partition to the aerosol phase. The threshold can be adjusted accordingly to fit any other objective, depending on the analysis required (e.g., a prescribed PM response). With these considerations, the threshold factors are 9 for both compounds and the characteristic pH values obtain the very simple formulations $pH' = -log[9\Psi W_i]$ for nitrate and $pH'' = log[9\Phi W_i]$ for ammonium. Apart from the value of the parameters $\Psi$, $\Phi$ (which vary mainly with T), pH' and

pH" vary only with $W_i$ – with a logarithmic dependence. Figure 2 displays their variation for 273



K (panel a) and 298 K (panel b). Nitrate tends to exhibit a decrease in *pH'* with increasing $W_i$, and vice-versa for ammonium and $pH"$.

Based on the values of the characteristic pH and its relation to the aerosol pH, we can then determine whether the aerosol responds to changes in nitrate or ammonium – as only when pH >

pH' (or pH < pH") does the aerosol become sensitive to changes in $NO_3^T$ (or $NH_3^T$). This realization constitutes the basis of our new framework and aerosol can belong to one of four distinct chemical regimes:

- Regime 1: Not sensitive to either NH₃ or HNO₃: this occurs when pH > pH" and pH < pH'.
  This regime is termed "*NH₃, HNO₃ insensitive*" or just "*Insensitive*".
- Regime 2: Not sensitive to NH₃; sensitive to HNO₃: this occurs when pH > pH" and
  pH > pH'. This regime is termed "*HNO₃ sensitive*".
- Regime 3: *Sensitive to both* NH₃ *and* HNO₃: this occurs when pH < pH" and pH < pH'
  pH > pH'. This regime is termed "*NH₃, HNO₃ sensitive*".
- Regime 4: *Sensitive to* NH₃ *and not sensitive to* HNO₃: pH < pH" and pH < pH'
pH > pH'. This regime is termed "*NH₃ sensitive*".

Figure 3 indicates these four regions in white (Regime 1), blue (Regime 2), purple (Regime 3), and red (Regime 4) for 273 K (Figure 3a) and 298 K (Figure 3b). Therefore, any specific set of data (from observations or a model), based on its corresponding aerosol acidity and liquid water, places it on one of the 4 above domains - which in turn determines the "chemical regime" of

aerosol response to $NH_3^T$ and/or $NO_3^T$. What is surprising is the emergence of a region of conditions where aerosol is insensitive to either NH₃ or HNO₃ – which occupies an increasingly large pH-LWC domain as the temperature increases (Fig.3).

A characteristic point on the chemical regime map corresponds to where the two lines "crossover", thus separating Regime 1 from Regime 3, and Regime 2 from Regime 4. This "critical" point

corresponds to a characteristic value of LWC, $W_i^*$, that is easily found by equating pH' with pH":

$$W_i^* = \left[\left(\frac{1-\alpha}{\alpha}\right)\left(\frac{1-\beta}{\beta}\right)\Phi\Psi\right]^{-1/2} \tag{7}$$

Substitution of $W_i^*$ into either Eq. 5 or 6 gives also the characteristic $pH^*$ of this crossover point:

$$pH^* = -\frac{1}{2}log\left(\frac{\Psi}{\Phi}\right) \tag{8}$$



Both $pH^*$, $W_i^*$ depend on temperature (Fig. 3). For T=298K and $\alpha = \beta = 0.1$, $\Psi \sim 7.38 \times 10^2$, $\Phi \sim 1.67 \times 10^7$ so $W_i^* \sim 3.5$ µg m$^{-3}$ and $pH^* \sim 2.2$. Therefore, for moderately acidic aerosol (pH*~2) and for moderate levels of liquid water content (a few µg m$^{-3}$ liquid water content) aerosol tends to be insensitive to emissions of either $NH_3^T$ and $NO_3^T$ precursors. For higher (or lower) pH levels, the aerosol transitions between regions 2 (or 4). For liquid water above $W_i^*$, there is a "transition pH" from an ammonia-sensitive to an exclusively nitrate-sensitive aerosol, which depends linearly on liquid water content (Fig. 3). Similarly, there is also another "transition pH" that defines when the aerosol becomes exclusively sensitive to $NH_3^T$ . Given the complexity of aerosol thermodynamics it is remarkable that such an apparently simple framework can be used to characterize the regions of aerosol sensitivity to $NH_3^T$ and $NO_3^T$ emissions, with "coordinates" being pH and liquid water. This is illustrated in the following section.

### 3. Application of framework

The above framework requires knowledge of aerosol pH and liquid water content, which can be routinely calculated by state-of-the-art atmospheric chemical transport models (e.g., CMAQ, CAMx) during the course of any simulation. Thermodynamic analysis of ambient aerosol and gas-phase data also provides aerosol pH and liquid water content, therefore the above framework can be used to characterize the chemical domain of ambient and simulated aerosol.

The applicability of the chemical domain approach is demonstrated by its application to ambient data. For this purpose, we have selected more than 7700 data points obtained from observations over 5 locations worldwide: Cabauw (CBW), Tianjin (TJN), California (CNX), SE US (SAS), and a wintertime NE USA (WIN) study (Table 1). Each dataset displays a broad range of acidity, temperature, relative humidity and has been thoroughly studied and evaluated for the applicability of thermodynamic analysis. Each data point corresponds roughly to a 1-hr measurement, meaning that the chemical domains examined correspond to effectively the instantaneous response of PM to ammonia and nitric acid availability. In addition to the major aerosol species ammonium/ammonia, sulfate, nitrate/nitric acid, the datasets also contain chloride/hydrochloric acid, sodium, calcium, potassium and magnesium (not shown in Table 1) which contribute to the pH and liquid water levels predicted. However, not all of the data provide size-dependent composition, so our analysis is limited, here, to looking at the bulk fine PM composition. The





range of $\varepsilon(NO_3^-)$ and $\varepsilon(NH_4^+)$ for all the data examined is presented in Figure 4. Noted on the figure are also indicative domains that correspond to Regions 1 to 4. It is clear that each dataset has distinct characteristics that provide insight on the expected sensitivity of PM to $NH_3$ and $HNO_3$ emissions – as low $\varepsilon(NO_3^-)$, $\varepsilon(NH_4^+)$ correspond to low sensitivity of PM to their respective

precursor emissions. It is unclear however, based on ε alone, where this (in)sensitivity originates from; strong/weak acidity, high/low liquid water content or high/low temperature. The latter is important, given that those parameters in models shape the local sensitivity profiles. Much of the data are found towards the extremes in the partitioning fraction scale, leaving the central part of the diagram sparsely populated. However, this does not mean that aerosols are limited by one

component or the other, as much of the data are found to be in the region sensitive to both.

Figure 5 presented the chemical domain classifications for each location.  These data sets are used to provide an example and may not apply to all locations in the region. For each subplot, the characteristic curves are calculated using the average temperature of the dataset (presented in Table 1). From each subplot it becomes clear that every location (CBW, TJN, CNX, SAS, WIN) belongs

almost exclusively to a characteristic domain for the duration of the measurements. Cabauw, for example, is characterized by high enough $NH_3$ so that aerosol is not sensitive to variations of it. Nitric acid, on the other hand, is by far a limiting factor in PM formation, hence CBW is in the $HNO_3$-dominated regime throughout the year. For similar reasons, Tianjin is also mostly in the $HNO_3$-dominated region, although a fraction of the data points lie in the combined $NH_3$-$HNO_3$

region owing to the slightly more acidic conditions compared to CBW. The Southeast US (SAS) is considerably more acidic, and with an order of magnitude less liquid water content compared to CBW and TJN; for these reasons it belongs to the $NH_3$-sensitive regime (i.e., there is little $NH_4NO_3$ present in summer – even if total nitrate availability may be high). The California dataset is quite interesting, being one that partly occupies the insensitive region and then transitions to the

combined $NH_3$-$HNO_3$ region; in this dataset, the combination of moderate $NH_3$ levels, NVCs from sea-salt (keep or delete) and temperature maintain acidity at levels that make aerosol sensitive to both $NH_3$ and $HNO_3$ variations. The wintertime eastern US dataset (WIN) corresponds to a broad region (aircraft data set), hence the data naturally occupies multiple domains. The lower temperatures, however, prohibit most of the data from occupying any of the insensitive region;

most of the data occupies the $NH_3$-sensitive regime, owing to the strong acidity and low liquid water. One remarkable point, however, is that regardless of location, the transition point between





NH$_3$-dominated and HNO$_3$-dominated sensitivity always occurs at a pH around 2 but at variable levels of liquid water content. The latter is important, as pH emerges as a required but not sufficient condition to determine the type of aerosol sensitivity; too little water (i.e., liquid water below the characteristic value $W_i^*$) and the aerosol can be insensitive to NH$_3$, even if the pH is as low as 2

(Figure 5a). In the case of Cabauw conditions (Figure 5a), where aerosol liquid water ranges from 7-15 μg m$^{-3}$, the "transition pH" from an aerosol that is exclusively sensitive to NO$_3^T$ precursor emissions to one that is sensitive to both NH$_3^T$ and NO$_3^T$ ranges from 2.8 and 3.2, which is in perfect agreement with the analysis of Guo et al. (2018). The additional insight that our framework shows is that the "transition pH" varies with temperature and logarithmically with aerosol liquid

water content, in response to emissions and diurnal/seasonal variability and climate change. This insight, not apparent in the analysis of Guo et al. (2018), demonstrates the power and flexibility of the new framework.

## 4. Conclusions

Here we present a simple yet powerful way to understand when concentrations of nitric acid

(HNO$_3$) and ammonia (NH$_3$) from anthropogenic and biogenic emissions can considerably modulate particulate matter (PM) concentrations worldwide. The conceptual framework explicitly considers acidity (pH), aerosol liquid water content and temperature as the main parameters controlling secondary inorganic PM sensitivity and identifies four policy-relevant regimes: *i*) NH$_3$-dominated, *ii*) HNO$_3$-dominated, *iii*) both NH$_3$ and HNO$_3$, and, *iv*) a previously unidentified

domain where neither NH$_3$ and HNO$_3$ are important for PM formation (but only nonvolatile precursors such as NVCs and sulfate). When this framework is applied to ambient measurements and predictions of PM and gaseous precursors, the "chemical regime" of PM sensitivity to emissions is directly determined, allowing novel insights and eventually an important tool to evaluate models. The framework can be used to identify regions or time periods where pH and

liquid water content prediction errors matter for PM sensitivity assessments. With this deeper understanding, aerosol pH and associated liquid water content naturally emerge as policy-relevant parameters that have not been explicitly explored until now.





**Acknowledgements**

This work was supported by the project PyroTRACH (ERC-2016-COG) funded from H2020-EU.1.1. - Excellent Science - European Research Council (ERC), project ID 726165. We also

acknowledge support from the U.S. EPA under grant R83588001. Its contents are solely the responsibility of the grantee and do not necessarily represent the official views of the supporting agencies. Further, the U.S. government does not endorse the purchase of any commercial products or services mentioned in the publication. We also thank Hongyu Guo and Guoliang Shi for providing access to the data used here.

**Code and Data availability**

User access to data used in this manuscript is described in the citations referenced for each dataset. They also may be obtained by request by AN. The ISORROPIA-II thermodynamic equilibrium code is available at http://isorropia.epfl.ch.

**Competing interests**

The authors declare that they have no conflicts of interest.

**Author contributions**

AN initiated the study, developed the framework, carried out analysis of the data and wrote the initial draft. All authors provided feedback on the analysis approach and extensively commented on the manuscript.

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



**Table 1.** Characteristics of the datasets used for determining the sensitivity to $NH_3$ and $HNO_3$ emissions. Shown is the average relative humidity (RH), Temperature (K), and the concentration of major aerosol precursors ($\mu g \, \mu^{-3}$), while in the respective standard deviation for each parameter is shown in parenthesis.

| Dataset ID, Location, (Reference) | RH (%) | T-range | Sulfate | Total Ammonium | Total Nitrate |
|---|---|---|---|---|---|
| TJN, Tianjin, China (Shi et al., 2019) | 56.6 (12.4) | 301.8 (2.79) | 21.46 (10.99) | 37.74 (7.68) | 18.12 (11.50) |
| CNX, Pasadena, CA, USA (Guo et al., 2017) | 71.3 (15.5) | 291.1 (4.26) | 2.86 (1.70) | 3.44 (1.81) | 10.23 (9.74) |
| CBW, Cabauw, Netherlands (Guo et al., 2018) | 78.2 (14.8) | 282.2 (7.3) | 1.92 (1.57) | 9.3 (6.8) | 4.1 (3.9) |
| WIN, Eastern USA (Guo et al., 2016) | 56.1 (18.9) | 270.8 (6.52) | 1.02 (0.08) | 0.53 (0.44) | 2.12 (2.08) |
| SAS, Centerville, AL, USA (Guo et. al, 2015) | 72.7 (17.4) | 297.9 (3.45) | 1.81 (1.18) | 0.78 (0.50) | 0.12 (0.15) |





**Figure Captions**

**Figure 1.** Particle phase fraction of total nitrate, $\varepsilon(NO_3^-)$ (blue curve) and total ammonium, $\varepsilon(NH_4^+)$ (red curve) versus pH for a temperature of 288 K and an aerosol liquid water content of (a) 10 µg m$^{-3}$, and, (b) 0.5 µg m$^{-3}$. The blue-color zone denotes where aerosol responds strongly (i.e. $\frac{\partial NO_3^-}{\partial NO_3^T} \sim$

1) to the amount of total nitrate, orange where NH$_3$ dominates (i.e. $\frac{\partial NH_4^+}{\partial NH_3^T} \sim 1$), purple where both NH$_3$ and HNO$_3$ changes affect PM concentrations, and white where aerosol is relatively insensitive to NH$_3$ and HNO$_3$ fluctuations. In defining the sensitivity domains, we have assumed that a partitioning fraction of 10% (black dotted lines), and its corresponding "characteristic" pH, defines where the aerosol becomes insensitive to changes in total NH$_3$, HNO$_3$.

**Figure 2.** Characteristic pH for defining when aerosol is sensitive to changes in available nitrate (blue lines) and ammonia (red lines) versus $W_i$. Results shown for a temperature of 298 K (dashed line) and 273 K (solid line). Note the relatively stronger effects of temperature changes on the characteristic pH for nitrate. Calculations carried out using the Excel sheet provided in the supplement.

**Figure 3.** Chemical domains of aerosol response to ammonia and nitrate emissions. Shown are results for 273 K (panel a) and 298 K (panel b). Note that there exists a fairly expansive region of acidity and liquid water content (especially for warmer temperatures) where aerosol is relatively insensitive to ammonia and nitrate emissions; here only non-volatiles (sulfate, NVCs) can have an appreciable impact on aerosol mass. Also important is the role of aerosol water in helping define

the chemical regime of aerosol sensitivity to precursors.

**Figure 4.** Aerosol partitioning fraction for total ammonia/ammonium and nitric acid/nitrate for the 5 regions examined: a) Cabauw - CBW, b) CalNex - CNX, c) Tianjin – TJN, d) SOAS – SAS, and, e) E. United States (WIN).

**Figure 5.** Chemical domains of sensitivity of aerosol to NH$_3$ and NO$_x$ emissions for 5 regions

examined: a) Cabauw - CBW, b) CalNex - CNX, c) Tianjin – TJN, d) SOAS – SAS, and, e) E. United States (WIN).





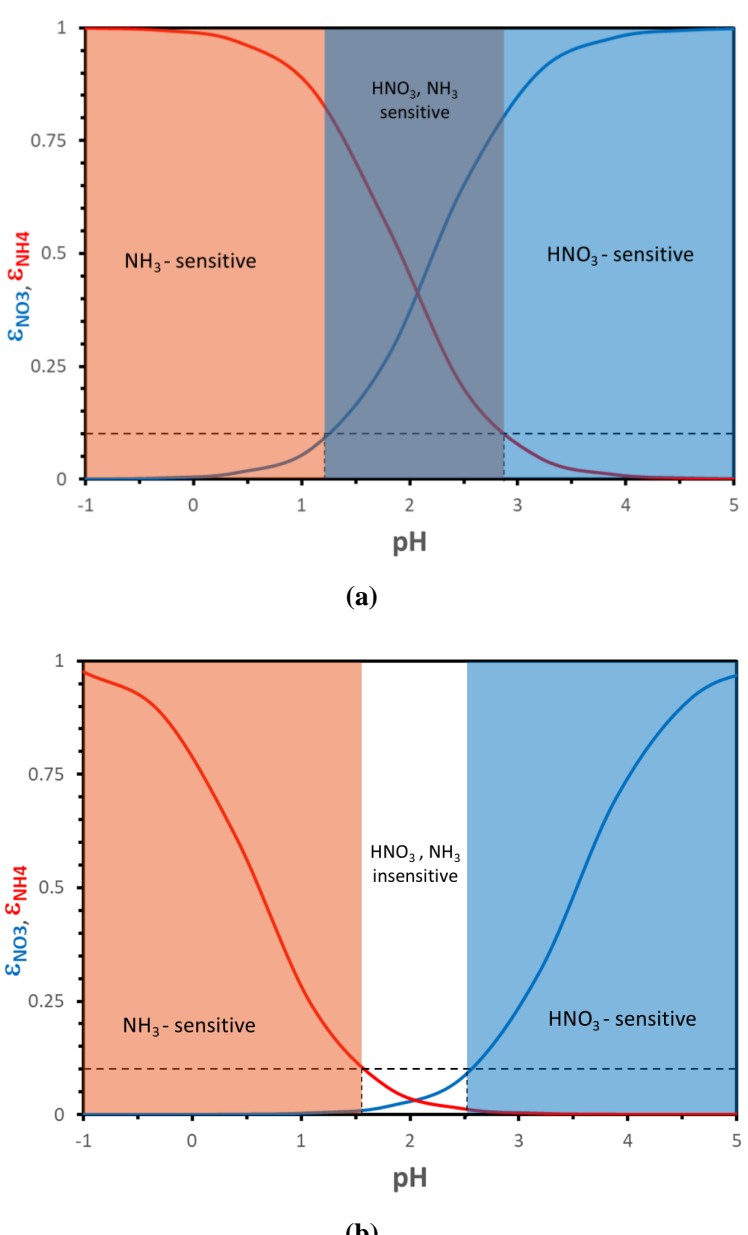

**(a)**

**(b)**

**Figure 1.**



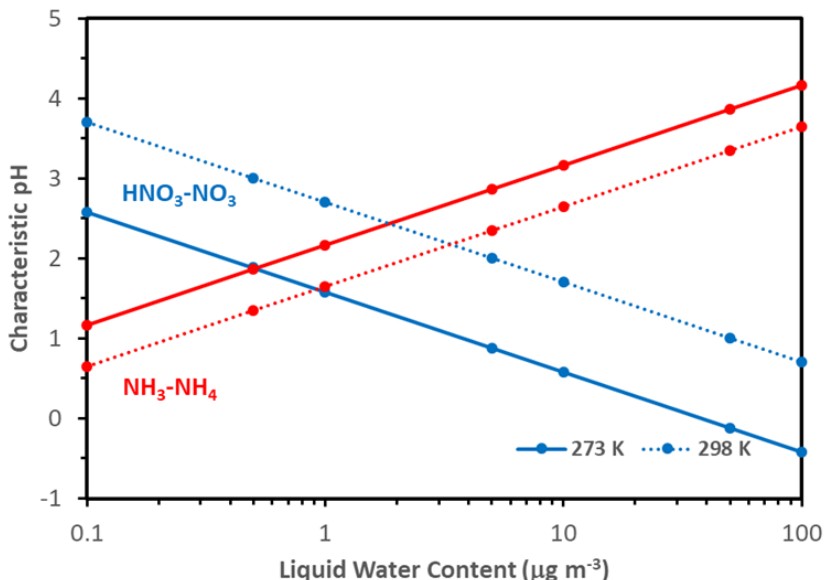

**Figure 2.**





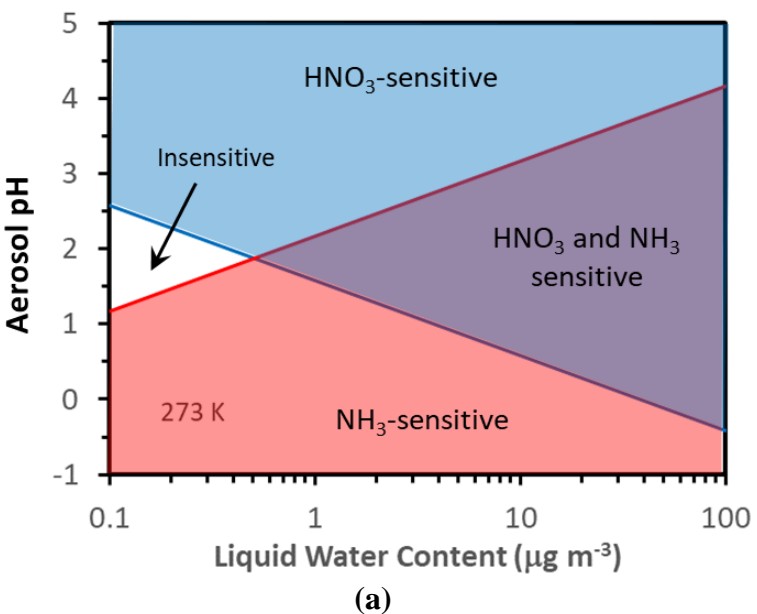

(a)

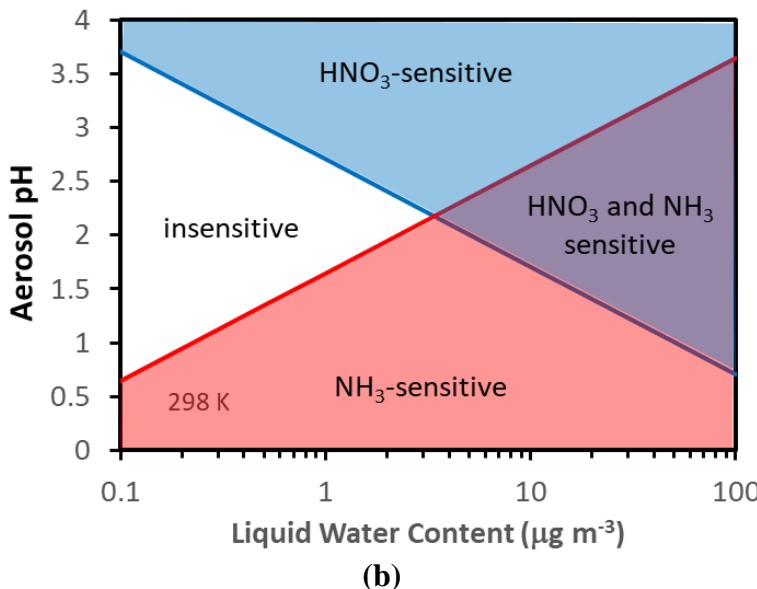

(b)

**Figure 3.**





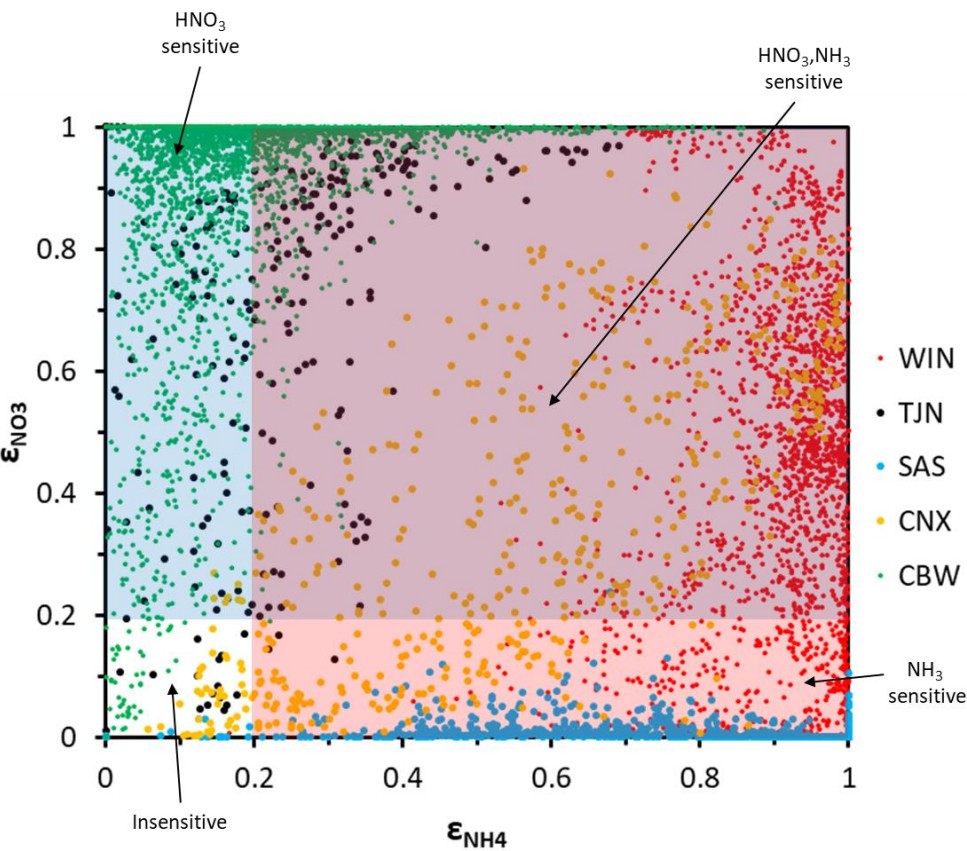

**Figure 4.**





**Figure 5.**