# Peer review of "Aerosol pH and liquid water content determine when particulate matter is sensitive to ammonia and nitrate availability"

_Atmospheric Chemistry and Physics, 2019_

## Referee Comment (RC1) · Anonymous Referee #1 · 13 Nov 2019

This is an excellent and important paper that introduces a new conceptual model for different aerosol "chemical regimes" based on a simple framework formalized around ammonia and nitric acid partitioning between the gas and aerosol phases as a function of aerosol pH and liquid water. The paper is well written and easy to follow, and provides a very useful way of quickly identifying the aerosol chemical regime, and more specifically the sensitivity of particulate mass to changes in gas phase ammonia and nitric acid emissions. This new framework will be extremely useful for efficiently employing models to evaluate expected changes to particle mass loadings given different emissions scenarios, which will be useful for informing policy makers.

[Figure]

One question I have is whether there might be other potential variables of importance (in addition to temperature, which the authors of course acknowledge)? For example, ionic strength, especially at very low LWC values.

I also wonder, when evaluating real ambient aerosols, how organic compounds could skew these results, especially since organics can comprise a substantial fraction of the aerosol mass (and therefore potentially impact both aerosol pH and liquid water). The authors do not discuss organic aerosol at all, which seems to be a big oversight. However, this closing sentence makes it clear that the authors are aware of potential biases like this, and that these potential biases do not detract from the basic goal of the framework, which includes identifying "...regions or time periods where pH and liquid water content prediction errors matter for PM sensitivity assessments."

While the different regimes are very useful in their own right, I was expecting that the lambda and kappa values described in the methods section would be used later in the paper to further characterize the aerosol mass sensitivity (e.g. not just *that* the aerosol is sensitive to NH4 emissions, but *how* sensitive). This might be a nice addition, to further show how the data could be used by policy makers.

———————————————— Minor comments: 'NVC' used but not defined in the abstract.

Why use "kappa" for the ratio of PM mass formed per mole of NO3, when kappa is already well-known and widely used by the aerosol community as the hygroscopicity parameter? Is there no other Greek letter available?

This statement is a bit open-ended: "...if nitric acid were replacing chloride in seasalt (e.g., conversion of NaCl to NaNO3), the ratio would be $\kappa$ = (85-58.4)/62=0.43" This seems to imply that the Cl (from NaCl) and H (from HNO3) evaporate from the aerosol as HCl. If that is the assumption, this should be stated explicitly. If volatilization of CO2 is assumed to occur during reaction between alkaline dust and HNO3, this should also be stated explicitly, for clarity.
The examples given for lambda are helpful, but more extreme examples are also possible (e.g. ammonium bioxalate, lambda = 142/14 = 10).

Typo in table 1 caption: (ug u-3)

In Fig. 4 it is unclear why the shaded regions extend to epsilon of 0.2 (rather than 0.1)

Line 26 on Page 9: "(keep or delete)" I suppose you should keep this statement ("NVCs from sea-salt"). Of course, the mixing state of the NVCs is central to their impact. This should probably be stated, succinctly, if the statement is indeed kept.

---

## Referee Comment (RC2) · Anonymous Referee #2 · 14 Nov 2019

The manuscript "Aerosol pH and liquid water content determine when particulate matter is sensitive to ammonia and nitrate availability" by Nenes and co-workers presents a thermodynamically consistent framework to assess the sensitivity of aerosol particulate matter (PM) to NH3 and HNO3 availability. The framework uses temperature, aerosol pH and liquid water content as main parameters to infer four regimes identified as 1) HNO3 sensitive (total nitrate reduction is effective to reducing PM); 2) NH3 sensitive (NH3 reduction is effective to reducing PM); 3) NH3 and HNO3 sensitive (both NOx and NH3 reductions are effective in reducing PM) and 4) a regime where neither NH3 nor HNO3 is important for PM levels. This latter regime is perhaps the most interesting aspect of the paper as, to my knowledge, it is the first time it has been pointed out.
The manuscript presents results that are of interest to the scientific community, it is well written and overall the concepts are expressed with clarity.

The main criticism I have is that while the authors explain well when this framework works well, they don't clarify when it is necessary to pay extra care at using it e.g., temperature below 273 K? Low aerosol water content? And in what instances aerosol pH and/or water content calculations are less reliable (as those quantities cannot be measured directly). Additionally, a mention on how the presence of an organic fraction in the aerosol mass could affect the pH and aerosol water content would help to guide the reader towards better use of this framework. I think that adding a sentence or two addressing the possible pitfalls that could occur in using this framework in the wrong domain would be of great help to the readers.

Minor comments

page 1 line 23: the four regimes are named in a different way here than in the main text. The naming used in the abstract is somehow confusing as "NH3-dominated" and "HNO3-dominated" does not immediately tell if the regime name refers to the aerosol phase, gas phase or total (aerosol+gas) therefore the reader could have a hard time to understand if the "NH3-dominated" regime is the same or the opposite than the "NH3 sensitive" regime. I recommend harmonizing the name of the four regimes and adopt in the abstract the same clearer nomenclature used at page7 lines 9 to 15 in the main text.

page 1 line 24: "... neither NH3 and HNO3 ..." maybe "... neither ... nor "?

page 6 line 3: "... the e(NO3-) is nearly 1 and almost all nitrate (NO3_T) is in the gas phase (HNO3)" This sentence is confusing. If the fraction of total nitrate in the aerosol phase is near 1 how can it be that almost all nitrate (NO3_T) is in the gas phase?

Table 1: indicating the units in the header (not only in the caption) would be useful to the reader

[Figure]

Figure 5: indicating the temperature ranges for each data set would be useful to the reader

[Figure]

---

## Author Comment (AC1) · 4 Jan 2020

**Response to Reviewer #1 comments:**

This is an excellent and important paper that introduces a new conceptual model for different aerosol "chemical regimes" based on a simple framework formalized around ammonia and nitric acid partitioning between the gas and aerosol phases as a function of aerosol pH and liquid water. The paper is well written and easy to follow, and provides a very useful way of quickly identifying the aerosol chemical regime, and more specifically, the sensitivity of particulate mass to changes in gas phase ammonia and nitric acid emissions. This new framework will be extremely useful for efficiently employing models to evaluate expected changes to particle mass loadings given different emissions scenarios, which will be useful for informing policy makers.

*We thank the reviewer for the enthusiastic response and thoughtful comments that have improved the manuscript. Below, we include the response to comments and questions raised.*

**Reviewer comment**: "One question I have is whether there might be other potential variables of importance (in addition to temperature, which the authors of course acknowledge)? For example, ionic strength, especially at very low LWC values."

**Answer:** *The reviewer raises a good point. Ionic strength is indeed a variable that affects the analysis and resulting maps – and is already considered by the framework in the Φ, Ψ terms. Given that the pH vs. W curves depend logarithmically on Φ, Ψ (hence activity coefficients), ionic strength impacts on the pH is in general of second order. For this reason, we consider the dataset-average activity coefficient when plotting the relevant curves.*

**Reviewer comment**: "I also wonder, when evaluating real ambient aerosols, how organic compounds could skew these results, especially since organics can comprise a substantial fraction of the aerosol mass (and therefore potentially impact both aerosol pH and liquid water). The authors do not discuss organic aerosol at all, which seems to be a big oversight. However, this closing sentence makes it clear that the authors are aware of potential biases like this, and that these potential biases do not detract from the basic goal of the framework, which includes identifying ": : :regions or time periods where pH and liquid water content prediction errors matter for PM sensitivity assessments.""

**Answer:** *This is an excellent point, and we should have discussed this more in the original manuscript. The effect of organic species – in terms of their impacts on liquid water and activity coefficients – is secondary, as long as the aerosol is a single phase (Battaglia et al., 2019). As long as the single phase requirement is satisfied, the framework will provide quite plausible results.*

**Reviewer comment**: "While the different regimes are very useful in their own right, I was expecting that the lambda and kappa values described in the methods section would be used later

in the paper to further characterize the aerosol mass sensitivity (e.g. not just *that* the aerosol is sensitive to NH4 emissions, but *how* sensitive). This might be a nice addition, to further show how the data could be used by policy makers."

*Answer: This is a very good suggestion. We have added such points in the revised text.*

**Reviewer comment**: "'NVC' used but not defined in the abstract."

*Answer: Thank you for pointing this out. It is now addressed.*

**Reviewer comment**: "Why use "kappa" for the ratio of PM mass formed per mole of NO3, when kappa is already well-known and widely used by the aerosol community as the hygroscopicity parameter? Is there no other Greek letter available?".

*Answer: Indeed so. We have replaced "kappa" with the Greek letter "zeta" ($\zeta$).*

**Reviewer comment**: "The examples given for lambda are helpful, but more extreme examples are also possible (e.g. ammonium bioxalate, lambda = 142/14 = 10)".

*Answer: The discussion in the original manuscript focused on the most likely values, given what we know about aerosol composition. However, we will note this extreme value, even if $\lambda=10$ assumes that oxalic acid resides completely in the gas phase before it condenses to the aerosol in the form of ammonium bioxalate; pure oxalic acid can reside substantially in the aerosol (~50%) regardless of pH (Nah et al., 2018), therefore $\lambda$ can be considerably less than 10.*

**Reviewer comment**: "Typo in table 1 caption: (ug u-3)".

*Answer: Corrected.*

**Reviewer comment**: "In Fig. 4 it is unclear why the shaded regions extend to epsilon of 0.2 (rather than 0.1)".

*Answer: Thank you for pointing this out. The issue is now corrected.*

**Reviewer comment**: "Line 26 on Page 9: "(keep or delete)" I suppose you should keep this statement ("NVCs from sea-salt"). Of course, the mixing state of the NVCs is central to their impact. This should probably be stated, succinctly, if the statement is indeed kept".

*Answer: Thank you for these suggestions. The manuscript has been modified as suggested.*

**Reference:**

Battaglia Jr., M. A., Weber, R. J., Nenes, A., and Hennigan, C. J.: Effects of water-soluble organic carbon on aerosol pH, Atmos. Chem. Phys., 19, 14607–14620, https://doi.org/10.5194/acp-19-14607-2019, 2019

Nah,T., Guo,H., Sullivan, A.P., Chen, Y., Tanner, D. J., Nenes, A., Russell, A., Ng, N. L., Huey, L. G. and R. J. Weber: Characterization of Aerosol Composition, Aerosol Acidity and Organic Acid Partitioning at an Agriculture-Intensive Rural Southeastern U.S. Site, Atmos.Chem.Phys. 18, 11471-11491, https://doi.org/10.5194/acp-18-11471-2018, 2018

---

## Author Comment (AC2) · 4 Jan 2020

**Response to Reviewer #2 comments:**

The manuscript "Aerosol pH and liquid water content determine when particulate matter is sensitive to ammonia and nitrate availability" by Nenes and co-workers presents … a regime where neither NH$_3$ nor HNO$_3$ is important for PM levels. This latter regime is perhaps the most interesting aspect of the paper as, to my knowledge, it is the first time it has been pointed out. The manuscript presents results that are of interest to the scientific community, it is well written and overall the concepts are expressed with clarity.

*We want to thank the reviewer for the positive and thoughtful comments that improve the manuscript. Below, we include the response to comments and questions raised and outline the changes made to the text.*

**Reviewer comment**: "The main criticism I have is that while the authors explain well when this framework works well, they don't clarify when it is necessary to pay extra care at using it e.g., temperature below 273 K? Low aerosol water content? And in what instances aerosol pH and/or water content calculations are less reliable (as those quantities cannot be measured directly)".

*Answer: These are all important points. The framework works well, as long as the assumption of thermodynamic equilibrium and a single aqueous phase provides a good representation of the aerosol. Typically this is associated with humidity above 40% and for temperatures about which mass transfer limitations are not severely limited by highly viscous or semi-solid aerosol. This is discussed extensively in Pye et al. (2019), Battaglia et al (2019) and others.*

**Reviewer comment**: "Additionally, a mention on how the presence of an organic fraction in the aerosol mass could affect the pH and aerosol water content would help to guide the reader towards better use of this framework. I think that adding a sentence or two addressing the possible pitfalls that could occur in using this framework in the wrong domain would be of great help to the readers".

*Answer: Thank you for pointing this out. As formulated here, the framework does not imply that the water is associated with the species considered (ammonium, nitrate) but rather it is treated as a variable; pH is also treated as a variable and can be modulated from organics, NVCs, halogen ions, sulfates, carbonates, and other species. The main requirement is that the aerosol is dominated by a single aqueous phase, as discussed in Battaglia et al. (2019).*

**Reviewer comment**: "page 1 line 23: the four regimes are named in a different way here than in the main text. The naming used in the abstract is somehow confusing as "NH3-dominated" and "HNO3-dominated" does not immediately tell if the regime name refers to the aerosol phase, gas phase or total (aerosol+gas) therefore the reader could have a hard time to understand if the "NH3-

dominated" regime is the same or the opposite than the "NH3 sensitive" regime. I recommend harmonizing the name of the four regimes and adopt in the abstract the same clearer nomenclature used at page7 lines 9 to 15 in the main text".

*Answer: Thank you for noting this. We have addressed this issue now.*

**Reviewer comment**: "page 1 line 24: "... neither NH3 and HNO3 ..." maybe "... neither ... nor "?".

*Answer: Done*

**Reviewer comment**: "page 6 line 3: "... the e($NO_3-$) is nearly 1 and almost all nitrate ($NO_{3\_T}$) is in the gas phase ($HNO_3$)" This sentence is confusing. If the fraction of total nitrate in the aerosol phase is near 1 how can it be that almost all nitrate ($NO_{3\_T}$) is in the gas phase?".

*Answer: Thank you or pointing this out. It was a typo, which is subsequently addressed.*

**Reviewer comment**: "Table 1: indicating the units in the header (not only in the caption) would be useful to the reader".

*Answer: Done*

**Reviewer comment**: "Figure 5: indicating the temperature ranges for each data set would be useful to the reader".

*Answer: We have noted in the caption that the average temperature, humidity and composition – along with their variances – are provided in Table 1.*

**Reference:**

Battaglia Jr., M. A., Weber, R. J., Nenes, A., and Hennigan, C. J.: Effects of water-soluble organic carbon on aerosol pH, Atmos. Chem. Phys., 19, 14607–14620, https://doi.org/10.5194/acp-19-14607-2019, 2019

Pye, H. O. T., Nenes, A., Alexander, B., Ault, A. P., Barth, M. C., Clegg, S. L., Collett Jr., J. L., Fahey, K. M., Hennigan, C. J., Herrmann, H., Kanakidou, M., Kelly, J. T., Ku, I.-T., McNeill, V. F., Riemer, N., Schaefer, T., Shi, G., Tilgner, A., Walker, J. T., Wang, T., Weber, R., Xing, J.,

Zaveri, R. A., and Zuend, A.: The Acidity of Atmospheric Particles and Clouds, Atmos. Chem.
Phys. Discuss., https://doi.org/10.5194/acp-2019-889, in review, 2019

---

## Author Response (AR1)

[revised manuscript text omitted]

**(a)**

[Figure]

**(b)**

**Figure 3.**

[Figure]

[Figure]

**Figure 4.**

[Figure]

**Figure 5.**